# Learning Nonparametric Volterra Kernels with Gaussian Processes

**Magnus Ross**
Department of Computer Science
University of Sheffield, UK
mross1@sheffield.ac.uk

**Michael T. Smith**
Department of Computer Science
University of Sheffield, UK
m.t.smith@sheffield.ac.uk

**Mauricio A. Álvarez**
Department of Computer Science
University of Sheffield, UK
mauricio.alvarez@sheffield.ac.uk

## Abstract

This paper introduces a method for the nonparametric Bayesian learning of nonlinear operators, through the use of the Volterra series with kernels represented using Gaussian processes (GPs), which we term the nonparametric Volterra kernels model (NVKM). When the input function to the operator is unobserved and has a GP prior, the NVKM constitutes a powerful method for both single and multiple output regression, and can be viewed as a nonlinear and nonparametric latent force model. When the input function is observed, the NVKM can be used to perform Bayesian system identification. We use recent advances in efficient sampling of explicit functions from GPs to map process realisations through the Volterra series without resorting to numerical integration, allowing scalability through doubly stochastic variational inference, and avoiding the need for Gaussian approximations of the output processes. We demonstrate the performance of the model for both multiple output regression and system identification using standard benchmarks.

## 1   Introduction

Gaussian processes (GPs) constitute a general method for placing prior distributions over functions, with the properties of samples from the distribution being controlled primarily by the form of the covariance function [22]. Process convolutions (PCs) are one powerful method for building such covariance functions [4, 13, 3]. In the PC framework, the function we wish to model is assumed to be generated by the application of some convolution operator to a base GP with some simple covariance, and since linear operators applied to GPs result in GPs, the result is another GP with a covariance we deem desirable. PCs allow models for multiple correlated output functions to be built with ease, by assuming each output is generated by a different operator applied to the same base function, or set of functions [30, 13].

The PC framework unifies a number of different ideas in the GP literature. Latent force models (LFMs) [1] use PCs to include physics based inductive biases in multiple output GP (MOGP) models by using the Green's function of a linear differential operator as the kernel of the convolution. This leads to the interpretation of each output as having been generated by inputting a random latent force into a linear system, with physical properties described by the differential operator. The Gaussian process convolution model (GPCM) of Tobar et al. [29] treats the convolution kernel itself as an unknown function to be inferred from data, and places a GP prior over it. Linear systems are entirely

described by their Green's function, so one can interpret the GPCM as a nonparametric, linear LFM, in which the form of the system itself is inferred from data.

In the physical world, nonlinear systems are the norm, and linearity is an approximation. As a consequence it is desirable when dealing with physical data to have models that can incorporate nonlinearity naturally. Often however, given a certain set of data, it is not clear exactly what form this nonlinearity takes, and so introducing specific parametric nonlinear operators can be overly restrictive. Álvarez et al. [2] present a model known as the nonlinear convolved MOGP (NCMOGP) which introduces nonlinearity to MOGPs via the Volterra series [8], a nonlinear series expansion used widely for systems identification, whose properties are controlled by a set of square integrable functions of increasing dimensionality known as the Volterra kernels (VKs). The NCMOGP assumes these functions are both separable and homogeneous, and of parametric Gaussian form, it also approximates the outputs as GPs in order to make inference tractable.

The present work introduces a new model which drops the separability and homogeneity assumptions on the VKs, allows their form to be learned directly from data, and makes no approximation on the distribution of the outputs. We refer to it as the nonparametric Volterra kernels model (NVKM). We develop a fast sampling method for the NVKM which leverages the recent results of Wilson et al. [35] on the sampling of explicit functions from GPs to analytically map function realisations through the Volterra series, avoiding the need for computationally expensive and inaccurate high dimensional numerical integration. Fast sampling allows for the application of doubly stochastic variational inference (DSVI) [28] for scalable learning.

The NVKM is well suited to both single and multiple output regression problems, and can be thought of as an extension of the GPCM to both nonlinear systems and multiple outputs. The NVKM can also be interpreted as a nonlinear LFM in which the operator is learned directly from data. We additionally present a variation of the NVKM that can be used for Bayesian systems identification, where the task is to learn operator mappings between observed input and output data, and show that it allows for considerably better quantification of uncertainty than competing methods which use recurrence [18].

## 2    Background

In this section we give a brief introduction to the mathematical background of PCs and the Volterra series.

**Process convolutions**    In the PC framework, the set of output functions $\{f_d(t)\}_{d=1}^D$, with $D$ being the number of outputs, is generated by the application of some set of linear operators, specifically convolution operators, to a latent function $u$ represented by a Gaussian process, $f_d(t) = \int_{\mathcal{T}} G_d(t - \tau)u(\tau)d\tau$, where $\mathcal{T}$ is the domain of integration, and the function $G_d$ is known variously as the convolutional kernel, smoothing kernel, impulse response or Green's function, depending on context. Assuming that the input $u$ is bounded, the function $G_d$ must be square integrable to ensure the output is finite. A linear operator acting on a GP produces another GP [22], and so we obtain $D$ distinct GPs. Since the latent function $u$ is shared across the outputs, these $D$ GPs are correlated, allowing joint variations to be captured, whilst the convolution with $G_d$ adapts $u$ to each output. Álvarez et al. [3] show that many MOGP models can be recast in terms of the PC framework by particular choices of $G_d$ and $u$. In LFMs, $G_d$ is taken to be the Green's function of some differential operator. We can then interpret each output as resulting from a shared random force being fed into a distinct linear system, represented by the differential operator. The smoothing kernels are usually taken to have parametric, often Gaussian form. Tobar et al. [29] use the PC framework for a single output, and make the smoothing kernel itself a GP, giving rise to the GPCM. Bruinsma [7] extends the GPCM to the multiple output case, although the model was not applied to data.

**Volterra Series**    The PC framework can be extended to represent a broader class of output functions by instead considering the outputs $\{f_d(t)\}_{d=1}^D$ as being the result of some nonlinear system, acting on the latent function $u$. The Volterra series is a series approximation for nonlinear systems that is widely used in the field of system identification [8]. It is given by,

$$f(t) = \sum_{c=1}^{C} \int_{\mathcal{T}} G_c(t - \tau_1, \ldots, t - \tau_c) \prod_{j=1}^{c} u(\tau_j)\mathrm{d}\tau_j, \tag{1}$$

where $f$ is the system output, $G_c$ is the $c^{\text{th}}$ order VK, $u$ is the system input, and $C$ is the order of the approximation. We can think of the Volterra series as an extension of the well known Taylor expansion, that allows $f$ to have memory of the past values of $u$, that is to say $f$ depends on $u$ at all values $t \in \mathcal{T}$. The Volterra series can approximate a broad class of nonlinear operators, however it is unable to represent certain properties of general nonlinear systems, for example chaos. Álvarez et al. [2] use Equation (1) to construct the NCMOGP, applying the Volterra series with $d$ distinct sets of VKs $\{G_{d,c}\}_{c=1}^{C}$ to a shared latent GP input $u$, to produce outputs $\{f_d(t)\}_{d=1}^{D}$. Since the Volterra series is a nonlinear operator, the output becomes an intractable, non-Gaussian process. The authors perform inference by approximating the outputs as GPs and using the first and second moments of the output process to form its mean and covariance function. To enable to computation of these moments, the authors restrict the set of VKs to those which are both separable and homogeneous, i.e. $G_{d,c}(t_1, \ldots, t_c) = \prod_{i=1}^{c} G_d(t_i)$. Additionally, since the moment computation requires analytically solving a number of non-trivial convolution integrals, the authors only consider a Gaussian form for the VKs.

## 3 The nonparametric Volterra kernels model

The NVKM relaxes the restrictions of separability and homogeneity which are placed on the VKs in the NCMOGP, and represents these kernels as independent GPs, allowing their form and uncertainty to be inferred directly from data. The generative process for the NVKM can be stated as

$$
\begin{aligned}
u(t) &\sim \mathcal{GP}(0, k^{(u)}(t, t')), \\
G_{d,c}(\mathbf{t}) &\sim \mathcal{GP}(0, k^{(G_{d,c})}(\mathbf{t}, \mathbf{t}')), \quad \mathbf{t} \in \mathbb{R}^c, \quad \forall c \in 1, \ldots, C, \quad \forall d \in 1, \ldots, D, \\
f_d(t) &= \sum_{c=1}^{C} \int_{-\infty}^{\infty} G_{d,c}(t - \tau_1, \ldots, t - \tau_c) \prod_{j=1}^{c} u(\tau_j) \mathrm{d}\tau_j,
\end{aligned}
\tag{2}
$$

where $k^{(u)}(t, t')$ is the covariance function for the input process, and $k^{(G_{d,c})}(\mathbf{t}, \mathbf{t}')$ is the covariance function for the $c^{\text{th}}$ VK of the $d^{\text{th}}$ output. We follow Tobar et al. [29] in using the decaying square exponential (DSE) covariance for the VKs, which is a modification to the ubiquitous square exponential (SE) covariance that ensures the samples have finite energy.[1] The DSE covariance has the form

$$
k_{DSE}(\mathbf{t}, \mathbf{t}') = \sigma^2 \exp(-\alpha(\|\mathbf{t}\|^2 + \|\mathbf{t}'\|^2) - \gamma\|\mathbf{t} - \mathbf{t}'\|^2),
\tag{3}
$$

where $\| \cdot \|$ is the or $\ell^2$ norm, $\sigma$ is the amplitude, $\alpha$ controls the rate at which the samples decay away from the origin, and $\gamma$ is related to the length scale $l$ of the samples by $\gamma = \frac{1}{l^2}$. A diagram of the generative process for the model is shown in Figure 1. Obtaining an exact distribution over the outputs $f_d$ is intractable, since it involves integration over nonlinear combinations of infinite dimensional stochastic processes. In order to sample from the model, and perform inference, approximations must be introduced. In particular, we employ the results of Wilson et al. [35] to sample in linear time, which enables efficient learning through the use of variational inducing points [27] with doubly stochastic variational inference (DSVI) [28].

### 3.1 Sampling

One could sample from the model by drawing from the input and filter GPs at some finite set of locations, and then using the samples with some method for numerical integration to find the output. As the dimensionality of the filters increases, however, many points would be needed to obtain an accurate answer, this quickly becomes computationally intractable, since sampling exactly from a GP has cubic time complexity with respect to the number of points requested. We can sidestep this problem, and avoid the need for any numerical integration, by representing samples from the GPs explicitly as functions. Using the results from Wilson et al. [35], and following their notation, we can write a sample from a GP $f : \mathbb{R}^c \to \mathbb{R}$, with covariance function $k(\mathbf{t}, \mathbf{t}')$, given $M$ inducing variables $\mathbf{v} \in \mathbb{R}^M$ with corresponding inducing inputs $\{\mathbf{z}_j\}_{j=1}^{M}$, with $\mathbf{z}_j \in \mathbb{R}^c$, as

$$
(f|\mathbf{v})(\mathbf{t}) = \sum_{i=1}^{N_b} w_i \phi_i(\mathbf{t}) + \sum_{j=1}^{M} q_j k(\mathbf{t}, \mathbf{z}_j),
\tag{4}
$$

---

[1]This is shown in Appendix A of the supplemental material.

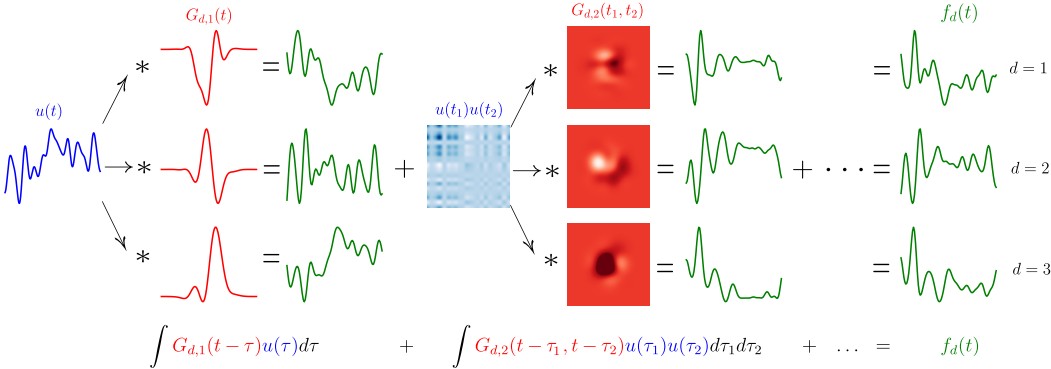

Figure 1: A diagram of the generative process for the NVKM with $C = 2$ and $D = 3$, showing the stages of computation of the first order (on the left side) and second order (in the center) terms of the Volterra series for three outputs, shown in the rows, with the star representing a convolution. The 1D contribution from the second order term is obtained by taking the diagonal of the result of the 2D convolution.

where $\{\phi_i\}_{i=1}^{N_b}$ is a random Fourier basis, with $N_b$ being the number of basis functions in the approximation, $\mathbf{w} \in \mathbb{R}^{N_b}$ with entries $w_i \sim \mathcal{N}(0, 1)$, and $q_j$ are the entries of the vector $\mathbf{q} = \mathbf{K}^{-1}(\mathbf{v} - \Phi\mathbf{w})$, where $\mathbf{K} \in \mathbb{R}^{M \times M}$, with elements $K_{m,n} = k(\mathbf{z}_n, \mathbf{z}_m)$, is the covariance matrix of the inducing points, and $\Phi \in \mathbb{R}^{M \times N_b}$ is a feature matrix, with each basis function being evaluated at each inducing location. The random Fourier basis is obtained by first sampling $\beta_i \sim U(0, 2\pi)$, where $U$ is the uniform distribution, and then sampling $\boldsymbol{\theta}_i \sim FT(k)$, where $FT$ is the Fourier transform of the covariance function, which is the spectral density of the process. The basis functions are then given by $\phi_i(\mathbf{t}) = \sqrt{2/N_b} \cos(\boldsymbol{\theta}_i^\top \mathbf{t} + \beta_i)$. We can see Equation (4) as consisting of an approximate GP prior, using a random Fourier features approximation [21], with a correction term which uses Matheron's update rule to account for the inducing points. By using Equation (4), samples can be obtained in linear time with respect to the number of requested points. It should be noted that Equation (4) only applies to GPs with stationary kernels, however, the DSE covariance required for $G_{c,d}$, is non-stationary. It can be shown that the process $\exp(-\alpha||\mathbf{t}||^2)G'_{c,d}(\mathbf{t})$ has the DSE covariance if $G'_{c,d}$ has the SE covariance. We can then write a sample from the output of the NVKM as,

$$(f_d | \{\mathbf{v}_{d,c}^G\}_{c=1}^C, \mathbf{v}^u)(t) = \\ \sum_{c=1}^C \int_{-\infty}^{\infty} e^{-\alpha \sum_{i=1}^c (t-\tau_i)^2} (G'_{d,c} | \mathbf{v}_{d,c}^G)(t - \tau_1, \dots, t - \tau_c) \prod_{j=1}^c (u|\mathbf{v}^u)(\tau_j) \mathrm{d}\tau_j, \quad (5)$$

which can be computed analytically, assuming that $u$ also has the SE covariance, by representing the Fourier basis in complex form, and factorising the integrals, leading to combinations of sums and products of single dimensional integrals of the form $\int_{-\infty}^{\infty} \exp(-ax^2 + bx) dx = \sqrt{\pi/a} \exp(b^2/4a)$. See Appendix B in the supplemental material for details of the computation.

## 3.2 Inference

Learning with the NVKM implies making inference of the input process $u$, along with all VKs $\{G_{d,c}\}_{c,d=1}^{C,D}$, from observed output data $\{\mathbf{y}_d\}_{d=1}^D$ with $\mathbf{y}_d \in \mathbb{R}^{N_d}$, which are the functions $\{f_d\}_{d=1}^D$ evaluated at points $\{\mathbf{t}_d\}_{d=1}^D$ with $\mathbf{t}_d \in \mathbb{R}^{N_d}$, corrupted by some i.i.d Gaussian noise. That is to say $y_{d,i} = f_d(t_{d,i}) + \epsilon_{d,i}$ with $\epsilon \sim \mathcal{N}(0, \sigma_{y_d}^2)$. Let $\mathbf{v}_{d,c}^G = G_{d,c}(\mathbf{z}_{d,c}^G)$ denote the inducing points for the VKs, and $\mathbf{v}^u = u(\mathbf{z}^u)$ denote the inducing points for the input. The joint distribution over these

inducing points and the latent functions then has the following form

$$p(\{\mathbf{y}_d\}_{d=1}^{D}, \{G_{d,c}, \mathbf{v}_{d,c}^{G}\}_{c,d=1}^{C,D}, u, \mathbf{v}^u) =$$
$$\prod_{d,i=1}^{D,N_d} p(y_{d,i}|f_d(t_{d,i})) \prod_{d,c=1}^{D,C} p(G_{d,c}|\mathbf{v}_{d,c}^G)p(\mathbf{v}_{d,c}^G)p(u|\mathbf{v}^u)p(\mathbf{v}^u), \quad (6)$$

where $f_d(t_{d,i})$ depends on the VKs and input through Equation (2), the likelihood is $p(y_{d,i}|f_d(t_{d,i})) = \mathcal{N}(y_{d,i}; f_d(t_{d,i}), \sigma_{y_d}^2)$ , $p(G_{d,c}|\mathbf{v}_{d,c}^G)$ and $p(u|\mathbf{v}^u)$ are GP posterior distributions, and $p(\mathbf{v}^u)$ and $p(\mathbf{v}_{d,c}^G)$ are the prior distributions over the inducing points. The dependency structure of the model is described in Figure 2. We form an approximate variational distribution, in a similar way to Tobar et al. [29], using a structured mean field approximation. That is to say, we mirror the form of the true joint distribution, and replace the prior distributions over the inducing points with variational distributions, $q(\mathbf{v}_{d,c}^G)$ and $q(\mathbf{v}^u)$, leading to the variational distribution

$$q(\{G_{d,c}, \mathbf{v}_{d,c}^{G}\}_{c,d=1}^{C,D}, u, \mathbf{v}^u) = \prod_{d,c=1}^{D,C} p(G_{d,c}|\mathbf{v}_{d,c}^G)q(\mathbf{v}_{d,c}^G)p(u|\mathbf{v}^u)q(\mathbf{v}^u). \quad (7)$$

Given the assumed factorisation of the variational distibution, the optimal form of the variational posteriors are multivariate Gaussians, $q(\mathbf{v}^u) = \mathcal{N}(\mathbf{v}^u; \boldsymbol{\mu}^u, \boldsymbol{\Sigma}^u)$ and $q(\mathbf{v}_{d,c}^G) = \mathcal{N}(\mathbf{v}_{d,c}^G; \boldsymbol{\mu}_{d,c}^G, \boldsymbol{\Sigma}_{d,c}^G)$, where the mean vectors and covariance matrices of these distributions are variational parameters. This form of variational approximation leads to a variational lower bound,

$$\mathcal{F} = \sum_{d,i=1}^{D,N_d} \mathbb{E}_q[\log p(y_{d,i}|f_d(t_{d,i}))] - \sum_{d,c=1}^{D,C} \mathrm{KL}[q(\mathbf{v}_{d,c}^G)||p(\mathbf{v}_{d,c}^G)] - \mathrm{KL}[q(\mathbf{v}^u)||p(\mathbf{v}^u)], \quad (8)$$

where $\mathrm{KL}[.||.]$ represents the Kullback-Liebler (KL) divergence, for details see Appendix C in the supplemental material. The expression above is optimised using gradient descent. The KL divergences have closed form. The derivation of the bound and KL divergences are given in the supplementary material. The expectation of the log likelihood of the outputs, given in the first term, is intractable, due to the nature of nonlinearity introduced by the Volterra series. We instead compute a stochastic estimate of the log likelihood by sampling from the model and using,

$$\mathbb{E}_q[\log p(y_{d,i}|f_d(t_{d,i}))] \approx \frac{1}{S} \sum_{s=1}^{S} \log p(y_{d,i}|(f_d|\mathbf{v}_{d,c}^G, \mathbf{v}^u)(t_{d,i})), \quad (9)$$

where $\mathbf{v}_{d,c}^G$ and $\mathbf{v}^u$ are first sampled from their respective variational distributions, and then used in Equation (5) to generate a sample from $f_d$. To make the inference scheme scalable, we compute the bound on randomly sub-sampled mini batches of the data set, which alone is known as stochastic variational inference [14]. When this source of stochasticity is combined with the stochastic estimate of the expected log likelihood, we have DSVI [28].

In the standard NVKM model, the input process $u$ is a latent function with no observed data associated with it. There are many situations in which, instead of learning a distribution over some output functions alone, we wish to learn an operator mapping between an input function and an output function, or functions. That is to say, in addition to observing data $\{\mathbf{y}_d\}_{d=1}^{D}$ we also observed the input process $u$ at locations $\mathbf{t}^x \in \mathbb{R}^{N_x}$ corrupted with i.i.d noise, which we denote

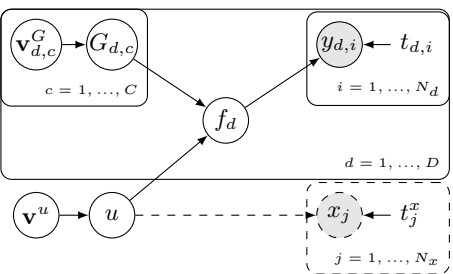

Figure 2: A graphical model for the NVKM, where the dashed elements are added to form the IO-NVKM. Note that nodes $u$, $f_d$, and $G_{d,c}$ are not random variables, they are random processes, but the distinction is not made in the diagram for the sake of clarity.

$\mathbf{x}$, so $x_i = u(t_i^x) + \epsilon_i^x$ with $\epsilon^x \sim \mathcal{N}(0, \sigma_x^2)$ . We apply a simple modification to the inference scheme of the NVKM, in order to form a new model which we term the input/output NVKM (IO-NVKM).

For the IO-NVKM, we pick up an additional likelihood term in Equation (6), with the variational distribution remaining unchanged, and we obtain a new bound,

$$\mathcal{F}_{IO} = \mathcal{F} + \sum_{j=1}^{N_x} \mathbb{E}_q[p(x_j|u(t_j^x))], \tag{10}$$

where $p(x_j|u(t_j^x)) = \mathcal{N}(x_j; u(t_j^x), \sigma_x^2)$, $\mathcal{F}$ is given by Equation (8) and we approximate the expectation as in Equation (9). The relationship between the NVKM and IO-NVKM is illustrated in Figure 2 where the dashed sections are added to form the IO model.

## 4    Related Work

In the this section, we give a brief overview of existing ideas in the literature that have some connection to the NVKM.

**Nonparametric covariances**    Covariance function design and selection is key to achieving good performance with GPs. Much work has been done on automatically determining covariance functions from data, for example by building complex covariances by composing simpler ones together [10], or by using deep neural networks to warp the input space in complex ways before the application of a simpler kernel [34]. Flexible parametric covariances can also be designed in frequency space [33]. Alternatively, some efforts have been made to learn covariance functions using GPs themselves. In addition the the GPCM, another model that learns covariances nonparametrically is due to Benton et al. [5], who use GPs to represent the log power spectral density, and then apply Bochner's theorem to convert this to a representation of the covariance function. GPs are fully specified by their first two moments, so by learning the covariance function and mean function, one knows all there is to know about the process. The present work uses the formalism of the Volterra series to learn the properties of more complex, non-Gaussian processes, nonparametrically. We can think of this as implicitly learning not just the first and second order moments of the process, but also the higher moments, depending on the value of $C$. In the case of $C = 1$ and $D = 1$, the NVKM and the GPCM are the same, except for the fact the GPCM uses the white noise as the input process, whereas we use an SE GP.

**LFMs and MOGPs**    As discussed in Section 2, we can interpret the first order filter function as the Green's function of some linear operator or system, and so by placing a GP prior over it, we implicitly place a prior over some set of linear systems [29]. Since standard LFMs use an operator of fixed form, we can interpret the NVKM in the case $C = 1$, $D \geq 1$ as being an LFM in which the generating differential equation itself is learned from data. LFMs can be extended to cases in which the differential operator is nonlinear. Hartikainen et al. [12] recast a specific nonlinear LFM in terms of a state space formalism allowing for inference in linear time. Lawrence et al. [17] use Markov chain Monte Carlo to infer the parameters of a specific nonlinear ODE describing gene regulation, using a GP as a latent input function. Ward et al. [31] use black box VI with inverse auto-regressive flows to infer parameters of the same ODE. When $C > 1$, the NVKM can be interpreted as a nonlinear nonparametric LFM. Álvarez et al. [2] use the fixed, parametric VKs to build an MOGP model. In contrast to the NVKM, they approximate the outputs as GPs and use analytical expressions for the moments to perform exact GP inference.

**Nonlinear system identification**    The IO-NVKM falls into the class of models which aim to perform system identification. A key concern of systems identification is determining how the output of certain systems, often represented by differential operators, respond to a given input. GPs have long been used for the identification of both linear and nonlinear systems. Many models exist which use GPs to recurrently map between previous and future states, including GP-NARX models [16], various state space models [26, 25] and recurrent GPs (RGPs) [19]. The thesis of Mattos [18] gives a summary of these methods. Worden et al. [36] detail a method for the combination of the GP-NARX model with a mechanistic model based on the physical properties of the system under study, leading to improved performance over purely data driven approaches. The IO-NVKM differs from these models in that instead of learning a mapping from the state of the system at a given point to the next state, we use GPs to learn an operator that maps the whole input function to the whole output function.

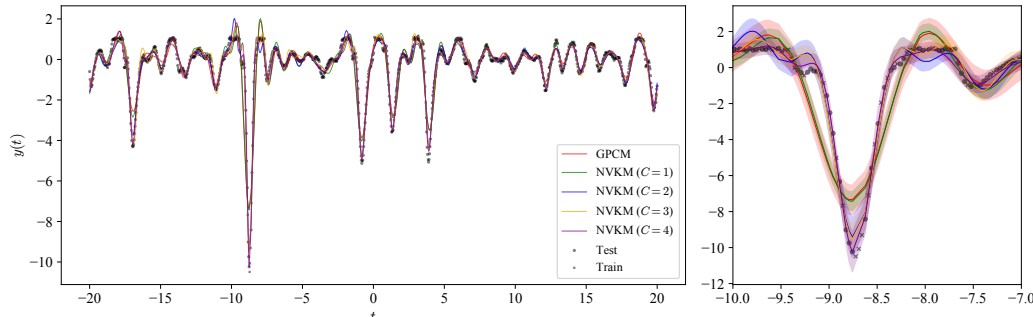

Figure 3: Model predictions on the synthetic data, with crosses indicating training points and dots indicating test points. The right plot shows an enhanced view of the peak around $t = -9$, and the shaded regions show $2\sigma$ confidence.

## 5 Experiments

For all the following experiments we place the inducing locations for both the input process and VKs on a fixed grid. The GP representing the $c^{\text{th}}$ VK has input dimension $c$, which means that the number of inducing points required to fully characterise it scales exponentially with $c$ if they are placed on a grid. For all experiments we use $15$, $10^2 = 100$, $6^3 = 216$ and $4^4 = 256$ inducing points, for each of the $1^{\text{st}}$ to $4^{\text{th}}$ order filters respectively, centered on zero. We treat the range of the points of each VK as a hyperparameter, and fix $\alpha$ such that the decaying part of the DSE covariance causes samples to be near zero at the edge of the range. For $u$ we use approximately 1/10 of the number of inducing points as data points (average number of data per output for multi-output problems). The VK GP length scales, VK GP amplitudes, and input GP amplitude are optimised along with the variational parameters by maximising the variational bound using gradient descent. For computational reasons, the input process length scale is fixed based on the spacing of the input inducing points, and the noise hyperparameters are fixed to a small value whilst the other hyperparameters and variational parameters are optimised, and then estimated afterwards by minimising the bound with all other variables fixed. The model is implemented using the Jax framework [6]. For all experiments we use Adam [15]. All models were trained on a single Nvidia K80 GPU. The code is available at `github.com/magnusross/nvkm`. In the following sections, details of the three main experiments are given. An additional experiment, in which data is sampled from the model, and the aim is to recover the VKs that generated it, is given in Appendix D of the supplemental material.

### 5.1 Synthetic data

To illustrate the advantage of including nonlinearity in the model we generate a synthetic single output regression problem which includes both hard and soft nonlinearities by sampling $g$ from an SE GP with length scale 2, computing $f_i(t) = \int e^{-2\tau^2} h_i(\tau) g(t - \tau) d\tau$ for $h_1(t) = \sin(6t)$, $h_2(t) = \sin^2(5t)$ and $h_3(t) = \cos(4t)$ by numerical integration, then computing the output as,

Table 1: Performance on the synthetic data set, showing mean and standard deviation for 10 repeats.

| Model | NMSE | NLPD |
|---|---|---|
| GPCM | 0.199 ±0.023 | 1.080 ±0.130 |
| NVKM ($C = 1$) | 0.196 ±0.047 | 2.084 ±0.398 |
| NVKM ($C = 2$) | 0.108 ±0.065 | 0.638 ±0.580 |
| NVKM ($C = 3$) | **0.055 ±0.016** | **0.124 ±0.107** |
| NVKM ($C = 4$) | 0.084 ±0.087 | 0.149 ±0.331 |

$$y(t) = \min(5f_1(t)f_2(t) + 5f_3^3(t), 1) + \epsilon, \tag{11}$$

with $\epsilon \sim \mathcal{N}(0, 0.05^2)$. We generate 1200 points in the range $t = [-20, 20]$ and use a random subset of a third for training and the rest for testing. Table 1 shows the normalised mean square errors (NMSEs) and negative log probability densities (NLPDs) on the test set for the NVKM with various values of $C$ as well as the GPCM, with repeats using a different random train/test split, and different random seeds.[2] As we would expect, the NMSE values are very similar for the NVKM

---

[2]Results generated using the implementation available at `github.com/wesselb/gpcm`

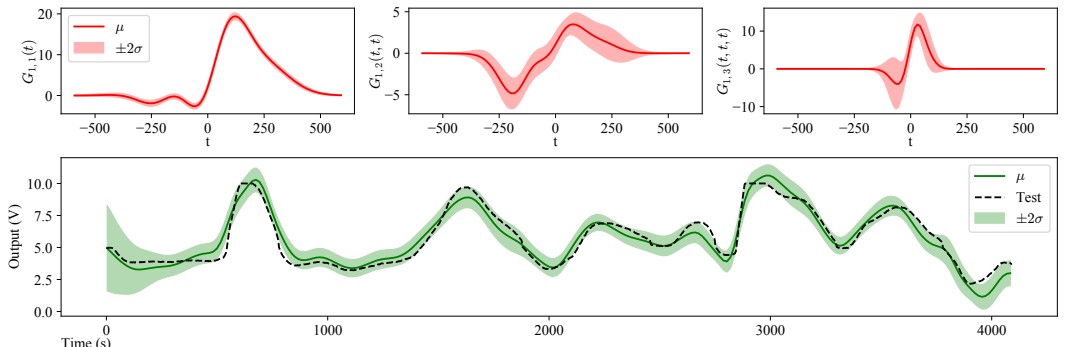

Figure 4: (Top) Diagonal of the inferred Volterra kernels for the the IO-NVKM with $C = 3$, showing $2\sigma$ confidence region. (Bottom) The predicted output for the test set, with the dashed line showing the true values.

with $C = 1$ and the GPCM, since the models are nearly equivalent except for the prior on the input GP. Interestingly the NLPD values are better for the GPCM than the NVKM with $C = 1$, likely due to the fact we do not optimise the noise jointly with the bound. As $C$ increases the performance of the NVKM improves until $C = 4$. The fact performance does not improve after $C = 3$ illustrates the difficulty of identifying higher order nonlinearities in a relatively small training set, an effect supported by the results of the *Cascaded Tanks* experiment in the following section. Although the $C = 4$ model does have more capacity to represent nonlinearities, the optimisation procedure is challenging, illustrated by the high variance of the results. Plots of the predictions for the model can be seen in Figure 3. We can see that increasing the nonlinearity for the NVKMs allows the sharp spike and the finer grained features, as well as the hard nonlinearities, to be captured simultaneously.

## 5.2 Cascaded tanks

To demonstrate the IO-NVKM, we use a standard benchmark for nonlinear systems identification know as *Cascaded Tanks* [24].[3] The system comprises two vertically stacked tanks filled with water, with water being pumped from a reservoir to the top tank, which then drains into the lower tank and finally back to the reservoir. The training data is two time series of 1024 points, one being the input to the system, which is the voltage fed into the pump, and the second being the output, which is the measured water level in the lower tank. For testing, an additional input signal, again of 1024 points, is provided, and the task is to predict the corresponding output water level. The system is considered challenging because it contains hard nonlinearities when the tanks reach maximum capacity and overflow (see the regions around 600s and 2900s in Figure 4), it

Table 2: Comparison of performance on the *Cascaded Tanks* dataset, with the last best models reported in [18]. $H$ indicates the number of hidden layers in the RGP.

| Model | RMSE | NLPD |
|---|---|---|
| IO-NVKM ($C = 1$) | 0.835 | 1.724 |
| IO-NVKM ($C = 2$) | 0.716 | 1.311 |
| IO-NVKM ($C = 3$) | 0.532 | **0.879** |
| IO-NVKM ($C = 4$) | 0.600 | 0.998 |
| RGP ($H = 1$) | 0.797 | 2.33 |
| RGP ($H = 2$) | **0.308** | 7.79 |
| GP-NARX | 1.50 | 1080 |
| Var. GP-NARX | 0.504 | 119.3 |

has unobserved internal state, and has a relatively small training set. Table 2 shows the predictive root mean square errors (RMSEs) and NLPDs for the IO-NVKM with various $C$, as well as four other GP based models for system identification from [18]. For each $C$, five random settings of VK ranges were tested, and each training was repeated three times with different initialisations. The setting and initialisation with the lowest combined NLPD on the training input and output data is shown. Although the RGP with $H = 2$ provides the best RMSE of the model, this comes at the cost of poor NLPD values. All IO-NVKMs achieve considerably better NLPD values than the alternatives indicating much better quantification of uncertainty. Of the IO-NVKMs, $C = 3$ performs best in both metrics. Figure 4 show the predictions of the $C = 3$ model on the test set, as well as the inferred VKs. The uncertainty in the VKs increases with their order, which is natural given the difficulty of estimating higher order nonlinear effects from a small training set. We can see the first order

---

[3] Available at `sites.google.com/view/nonlinear-benchmark/`

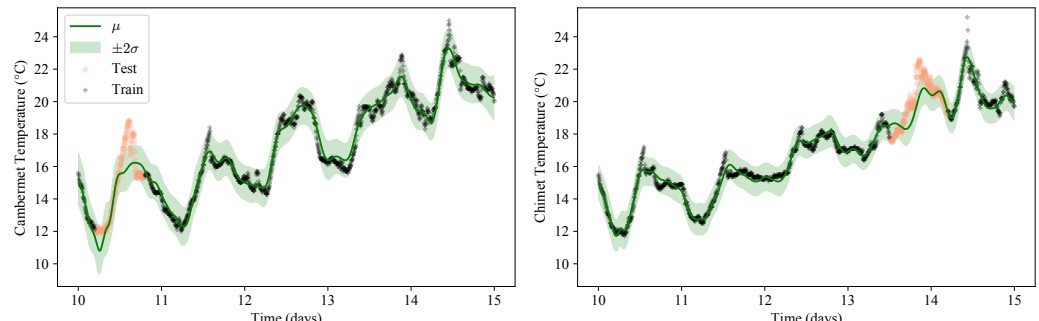

Figure 5: Predictive means and $2\sigma$ confidence regions from the NVKM with $C = 3$, on the Cambermet (left) and Chimet (right) outputs from the *Weather* data set. Orange circles are the artificially removed test data, black pluses are training data.

Table 3: Comparison of performance on the *Weather* data set, for the NVKM mean and standard deviation of three initialisation is shown, along with the best model from [2]

| Model | Cambermet | | Chimet | |
|---|---|---|---|---|
| | NMSE | NLPD | NMSE | NLPD |
| NVKM ($C = 1$) | **0.212±0.085** | **2.182±0.743** | 1.669±0.052 | 7.148±0.111 |
| NVKM ($C = 2$) | 0.440±0.286 | 3.884±2.380 | 0.939±0.216 | 4.143±1.197 |
| NVKM ($C = 3$) | 0.253±0.002 | 2.390±0.123 | 0.871±0.394 | 3.994±1.924 |
| NCMOGP ($C = 3$) | 0.44 | 2.33 | **0.43** | **2.18** |

kernel as approximating the response of a linearized system, and the third order allowing the sharp nonlinearities around 600s and 2900s to be captured. It should be noted that Worden et al. [36] achieve a much lower RMSE of 0.191 by using a specific physics model of the system in tandem with a GP-NARX model, but since we are considering purely data driven approaches here, it is not directly comparable.

## 5.3 Weather data

To illustrate the utility of the NVKM for multiple output regression problems, we consider a popular benchmark in the MOGP literature, consisting of multiple correlated time series of air temperature measurements taken at four nearby locations on the south coast of England, originally described by Nguyen et al. [20], which we refer to as *Weather*.[4] The four series are named Bramblemet, Sotonmet, Cambermet and Chimet, with 1425, 1097, 1441, and 1436 data points, respectively. Bramblemet and Sotonmet both contain regions of truly missing data, 173 and 201 points in a continuous region are artificially removed form Cambermet and Chimet with the task being to predict them based on the all the other data. Table 3 shows the performance of the multiple output NVKM on the *Weather* dataset, along with the best performing NCMOGP model of Álvarez et al. [2]. For each $C$, five random settings of VK ranges were tested, with each training being repeated three times with different initialisations, the setting with the best average NLPD value on the training data is shown, along with its standard deviation. All NVKM models show better or equivalent performance than the NCMOGP on the Cambermet output, but all show worse performance on the Chimet output, although on the Chimet output the variance between repeats is high. It should be noted that the LFM reported by Guarnizo and Álvarez [11] achieves much lower scores, having NMSEs of 0.11 and 0.19 on Cambermet and Chimet respectively, but that model uses six latent functions as opposed to a single latent function for the NVKM and NCMOGP. Including multiple latent functions may lead to large performance improvements for the NVKM and is a promising direction for future work.

---

[4]Available for download in a convenient from using the `wbml` package, `github.com/wesselb/wbml`

# 6 Discussion

**Societal Impacts**    It is possible that the NVKM model could incur some negative societal impacts. GP and MOGP models have long been applied to problems in robotics [9, 32]. Better inclusion of nonlinearities in these models may enhance the ability of robots, potentially leading to loss of jobs and livelihoods to automation. On the other hand, the NVKM may have a positive impact on society, its ability to model weather and climate data have been demonstrated in Section 5, and accurate climate forecasting models will likely play a key role in any solution to the ongoing climate crisis [23].

**Future Work**    There are a number of extensions to both the NVKM and IO-NVKM that could lead to substantial improvements in performance. As briefly mentioned in Section 5, the number of inducing points required for the VKs scales exponentially with the order of the series, meaning it is difficult to represent complex features in the higher order terms, without using a computationally intractable number of points. Whilst initially we saw the increased flexibility of non-separable VKs as a virtue, it may be that introducing separability leads to more powerful models, since the number of points needed to specify separable VKs scales linearly. Currently the models do not support multidimensional inputs, but this could be easily added, requiring the computation of a few extra integrals, with the complexity scaling linearly with the number of inputs. For the multiple output model, allowing a shared set of latent functions, with the input to each output's Volterra series being a trainable linear combination, in a similar way to LFMs, is highly likely to improve performance especially for problems with a large number of outputs. Additionally it is likely that the inference method can be improved significantly, perhaps by going beyond the structured mean field approximation and using a more flexible variational distribution, or even by forgoing VI entirely and sampling from the posterior using Markov chain Monte Carlo methods.

**Conclusions**    We have presented a novel model which uses Gaussian processes to learn the kernels of the Volterra series nonparametrically, allowing for the effective modeling of data with nonlinear properties. We have developed fast and scalable sampling and inference methods for the the model and shown its performance on single and multiple output regression problems. Additionally, a modification to the model was presented that achieves significantly better uncertainty quantification than competitors on a challenging benchmark for nonlinear systems identification.

## Acknowledgments and Disclosure of Funding

We thank Lizzy Cross, Felipe Tobar, and Carl Henrik Ek for helpful conversations. We would also like to thank Wessel Bruinsma for useful and insightful discussions and advice, as well as support in using the code for the GPCM. Magnus Ross and Michael T. Smith thank the Department of Computer Science at the University of Sheffield financial support. Mauricio A. Álvarez has been financed by the EPSRC Research Projects EP/R034303/1, EP/T00343X/2 and EP/V029045/1.

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
