# Learning Nonparametric Volterra Kernels with Gaussian Processes (Appendix)

## A Finite power of the outputs

We wish to show that the outputs, $f_d(t)$, of the NVKM have finite power, that is to say,

$$\mathbb{E}[f_d(t)^2] < \infty.$$

Since we are using the truncated Volterra series, with a finite number of terms, then it suffices to show that the $c$-th term has finite power. Using the slightly more concise notation $\boldsymbol{\tau} = (\tau_1, \ldots, \tau_c)$ and $t - \boldsymbol{\tau} = (t - \tau_1, \ldots, t - \tau_c)$, we have that

$$\mathbb{E}[f_{d,c}(t)^2] = \mathbb{E}\Big[\Big(\int_{-\infty}^{\infty} G_{d,c}(t - \boldsymbol{\tau}) \prod_{j=1}^{c} u(\tau_j) d\tau_j\Big)^2\Big]$$

$$= \mathbb{E}\Big[\int_{-\infty}^{\infty} G_{d,c}(t - \boldsymbol{\tau}) G_{d,c}(t - \boldsymbol{\tau}') \prod_{j=1}^{c} u(\tau_j) u(\tau_j') d\tau_j d\tau_j'\Big]$$

$$= \int_{-\infty}^{\infty} \mathbb{E}[G_{d,c}(t - \boldsymbol{\tau}) G_{d,c}(t - \boldsymbol{\tau}')] \mathbb{E}[\prod_{j=1}^{c} u(\tau_j) u(\tau_j')] d\tau_j d\tau_j'$$

$$= \int_{-\infty}^{\infty} k_{DSE}(t - \boldsymbol{\tau}, t - \boldsymbol{\tau}') \mathbb{E}[\prod_{j=1}^{c} u(\tau_j) u(\tau_j')] d\tau_j d\tau_j'$$

where we have use the independence of input and VKs on the third line, and the fact that VKs have zero mean on the fourth. Using the results of Song and Lee [3] we can express the product of Gaussian processes in terms of products and sums of their covariance functions, which means that, since we are using the SE covariance for $u$, which is bounded, the term $\mathbb{E}[\prod_{j=1}^{c} u(\tau_j) u(\tau_j')]$ is bounded. The exact form of the term is not given here due to its complexity, but details can be found in [1]. Letting $|\mathbb{E}[\prod_{j=1}^{c} u(\tau_j) u(\tau_j')]| = B_u$, we have that,

$$\mathbb{E}[f_{d,c}(t)^2] < B_u \int_{-\infty}^{\infty} k_{DSE}(t - \boldsymbol{\tau}, t - \boldsymbol{\tau}') d\boldsymbol{\tau} d\boldsymbol{\tau}'$$

$$= B_u \int_{-\infty}^{\infty} e^{-\alpha(\|t-\boldsymbol{\tau}\|^2 + \|t-\boldsymbol{\tau}'\|^2) - \gamma\|\boldsymbol{\tau}'-\boldsymbol{\tau}\|^2} d\boldsymbol{\tau} d\boldsymbol{\tau}'$$

$$= B_u \int_{-\infty}^{\infty} e^{-\alpha(\|t-\boldsymbol{\tau}\|^2 + \|t-\boldsymbol{\tau}'\|^2)} e^{-\gamma\|\boldsymbol{\tau}'-\boldsymbol{\tau}\|^2} d\boldsymbol{\tau} d\boldsymbol{\tau}'$$

$$< B_u \int_{-\infty}^{\infty} e^{-\alpha(\|t-\boldsymbol{\tau}\|^2 + \|t-\boldsymbol{\tau}'\|^2)} d\boldsymbol{\tau} d\boldsymbol{\tau}'$$

$$= B_u \Big(\frac{\pi}{\alpha}\Big)^c$$

where we have used that $|e^{-\gamma\|\boldsymbol{t}'-\boldsymbol{t}\|^2}| = 1$ on the fourth line. This implies the signal power is finite.

35th Conference on Neural Information Processing Systems (NeurIPS 2021).

## B Derivation of explicit sampling equations

Recall that we wish to analytically compute the integral in Equation (5), which was given by

$$(f_d|\{\mathbf{v}_{d,c}^G\}_{c=1}^C, \mathbf{v}^u)(t) = \sum_{c=1}^C \int_{-\infty}^\infty e^{-\alpha \sum_{i=1}^c (t-\tau_i)^2} (G_{d,c}'|\mathbf{v}_{d,c}^G)(t-\tau_1,\ldots,t-\tau_c) \prod_{j=1}^c (u|\mathbf{v}^u)(\tau_j) d\tau_j,$$

with all the processes $\{G_{d,c}\}_{c,d=1}^{C,D}$ and $u$ having the SE covariances, given by $k_{SE}(t,t') = \sigma^2 \exp(-p\|t - t'\|^2)$, where $\sigma$ is the amplitude of the process, and $p$ is the precision, which is related to the length scale $l$ by $p = \frac{1}{2l^2}$. For notational simplicity we will compute the integral for the $c$-th term, and drop the subscripts on $G$, we then substitute Equation (4) for $G$, giving,

$$I_{c,d} = \int_{-\infty}^\infty e^{-\alpha \sum_{m=1}^c (t-\tau_m)^2} (G'|\mathbf{v}^G)(t - \tau_1,\ldots,t - \tau_c) \prod_{k=1}^c (u|\mathbf{v}^u)(\tau_k) d\tau_k$$

$$= \int_{-\infty}^\infty e^{-\alpha \sum_{m=1}^c (t-\tau_m)^2} \Big( \sum_{i=1}^{N_b} w_i^{(G)} \phi_i^{(G)}(t - \boldsymbol{\tau}) + \sum_{j=1}^{M^{(G)}} q_j^{(G)} k^{(G)}(t - \boldsymbol{\tau}, \mathbf{z}_j^{(G)}) \Big) \prod_{k=1}^c (u|\mathbf{v}^u)(\tau_k) d\tau_k$$

$$= \sum_{i=1}^{N_b} w_i^{(G)} \underbrace{\int_{-\infty}^\infty e^{-\alpha \sum_{m=1}^c (t-\tau_m)^2} \phi_i^{(G)}(t - \boldsymbol{\tau}) \prod_{k=1}^c (u|\mathbf{v}^u)(\tau_k) d\tau_k}_{I_1}$$

$$+ \sum_{j=1}^{M^{(G)}} q_j^{(G)} \underbrace{\int_{-\infty}^\infty e^{-\alpha \sum_{m=1}^c (t-\tau_m)^2} k^{(G)}(t - \boldsymbol{\tau}, \mathbf{z}_j^{(G)}) \prod_{k=1}^c (u|\mathbf{v}^u)(\tau_k) d\tau_k}_{I_2}$$

with $\boldsymbol{\tau} = (\tau_1,\ldots,\tau_c)$ and $t - \boldsymbol{\tau} = (t - \tau_1,\ldots,t - \tau_c)$, with superscripts indicating which of the VK or input process the symbols from Equation (4) are associated with. We can see that there are two separate integrals to deal with, $I_1$ and $I_2$.

### B.1 $I_1$

We have

$$I_1 = \int_{-\infty}^\infty e^{-\alpha \sum_{m=1}^c (t-\tau_m)^2} \phi_i^{(G)}(t - \boldsymbol{\tau}) \prod_{k=1}^c (u|\mathbf{v}^u)(\tau_k) d\tau_k$$

$$= \sqrt{\frac{2}{N_b}} \int_{-\infty}^\infty e^{-\alpha \sum_{m=1}^c (t-\tau_m)^2} \cos(\sum_{j=1}^c \theta_{i,j}(t - \tau_j) + \beta_i) \prod_{k=1}^c (u|\mathbf{v}^u)(\tau_k) d\tau_k$$

$$= \frac{1}{2}\sqrt{\frac{2}{N_b}} \int_{-\infty}^\infty e^{-\alpha \sum_{m=1}^c (t-\tau_m)^2} e^{i(\sum_{j=1}^c \theta_{i,j}(t-\tau_j) + \beta_i)} \prod_{k=1}^c (u|\mathbf{v}^u)(\tau_k) d\tau_k$$

$$+ \frac{1}{2}\sqrt{\frac{2}{N_b}} \int_{-\infty}^\infty e^{-\alpha \sum_{m=1}^c (t-\tau_m)^2} e^{-i(\sum_{j=1}^c \theta_{i,j}(t-\tau_j) + \beta_i)} \prod_{k=1}^c (u|\mathbf{v}^u)(\tau_k) d\tau_k$$

$$= \frac{1}{2}\sqrt{\frac{2}{N_b}} e^{i\beta_i} \int_{-\infty}^\infty \prod_{j=1}^c e^{-\alpha(t-\tau_j)^2 + i\theta_{i,j}(t-\tau_j)} \prod_{k=1}^c (u|\mathbf{v}^u)(\tau_k) d\tau_k$$

$$+ \frac{1}{2}\sqrt{\frac{2}{N_b}} e^{-i\beta_i} \int_{-\infty}^\infty \prod_{j=1}^c e^{-\alpha(t-\tau_j)^2 - i\theta_{i,j}(t-\tau_j)} \prod_{k=1}^c (u|\mathbf{v}^u)(\tau_k) d\tau_k$$

$$= \frac{1}{2}\sqrt{\frac{2}{N_b}} e^{i\beta_i} \prod_{j=1}^c \int_{-\infty}^\infty e^{-\alpha(t-\tau)^2 + i\theta_{i,j}(t-\tau)} (u|\mathbf{v}^u)(\tau) d\tau$$

$$+ \frac{1}{2}\sqrt{\frac{2}{N_b}} e^{-i\beta_i} \prod_{j=1}^c \int_{-\infty}^\infty e^{-\alpha(t-\tau)^2 - i\theta_{i,j}(t-\tau)} (u|\mathbf{v}^u)(\tau) d\tau,$$

so we need to evaluate

$$\int_{-\infty}^{\infty} e^{-\alpha(t-\tau)^2 \pm i\theta_{i,j}^{(G)}(t-\tau)} (u|\mathbf{v}^u)(\tau)\mathrm{d}\tau$$

$$= \int_{-\infty}^{\infty} e^{-\alpha(t-\tau)^2 \pm i\theta_{i,j}^{(G)}(t-\tau)} \Big( \sum_{m=1}^{N_b} w_m^{(u)} \phi_m^{(u)}(\tau) + \sum_{n=1}^{M^{(u)}} q_n^{(u)} k^{(u)}(\tau, z_n^{(u)}) \Big)\mathrm{d}\tau$$

$$= \sqrt{\frac{2}{N_b}} \sum_{m=1}^{N_b} w_m^{(u)} \int_{-\infty}^{\infty} e^{-\alpha(t-\tau)^2 \pm i\theta_{i,j}^{(G)}(t-\tau)} \cos(\theta_m^{(u)}\tau + \beta_m^{(u)})d\tau$$

$$+ \sigma_u^2 \sum_{n=1}^{M^{(u)}} q_n^{(u)} \int_{-\infty}^{\infty} e^{-\alpha(t-\tau)^2 \pm i\theta_{i,j}^{(G)}(t-\tau)} e^{-p_u(\tau - z_n^{(u)})^2} d\tau$$

$$= \sqrt{\frac{2}{N_b}} \sum_{m=1}^{N_b} w_m^{(u)} I_{1a}(t; \alpha, \pm\theta_{i,j}^{(G)}, \theta_m^{(u)}, \beta_m^{(u)}) + \sigma_u^2 \sum_{n=1}^{M^{(u)}} q_n^{(u)} I_{1b}(t; \alpha, \pm\theta_{i,j}^{(G)}, p_u, z_n^{(u)}).$$

We can find explicit forms for $I_{1a}$,

$$I_{1a}(t; \alpha, \theta_1, \theta_2, \beta_2) = \int_{-\infty}^{\infty} e^{-\alpha(t-\tau)^2 + i\theta_1(t-\tau)} \cos(\theta_2\tau + \beta_2)d\tau$$

$$= \frac{\sqrt{\pi}}{2\sqrt{\alpha}} \left( 1 + e^{\frac{\theta_1\theta_2}{\alpha} + 2i\beta_2 + 2i\theta_2 t} \right) e^{-\frac{(\theta_1 + \theta_2)^2}{4\alpha} - i(\beta_2 + \theta_2 t)},$$

as well as for $I_{1b}$,

$$I_{1b}(t; \alpha, \theta_1, p_2, z_2) = \int_{-\infty}^{\infty} e^{-\alpha(t-\tau)^2 + i\theta_1(t-\tau)} e^{-p_2(\tau - z_2)^2} d\tau$$

$$= \frac{\sqrt{\pi}}{\sqrt{\alpha + p_2}} e^{\frac{-4\alpha p_2(t-z_2)^2 + i\theta_1(4p_2 t - 4p_2 z_2 + i\theta_1)}{4(\alpha + p_2)}}.$$

Putting it all together we get that $I_1$ is given by,

$$I_1 = \frac{1}{2}\sqrt{\frac{2}{N_b}} \Big( e^{i\beta_i^{(G)}} \prod_{j=1}^{c} \Big[ \sqrt{\frac{2}{N_b}} \sum_{m=1}^{N_b} w_m^{(u)} I_{1a}(t; \alpha, \theta_{i,j}^{(G)}, \theta_m^{(u)}, \beta_m^{(u)})$$

$$+ \sigma_u^2 \sum_{n=1}^{M^{(u)}} q_n^{(u)} I_{1b}(t; \alpha, \theta_{i,j}^{(G)}, p_u, z_n^{(u)}) \Big]$$

$$+ e^{-i\beta_i^{(G)}} \prod_{j=1}^{c} \Big[ \sqrt{\frac{2}{N_b}} \sum_{m=1}^{N_b} w_m^{(u)} I_{1a}(t; \alpha, -\theta_{i,j}^{(G)}, -\theta_m^{(u)}, \beta_m^{(u)})$$

$$+ \sigma_u^2 \sum_{n=1}^{M^{(u)}} q_n^{(u)} I_{1b}(t; \alpha, -\theta_{i,j}^{(G)}, -p_u, z_n^{(u)}) \Big] \Big)$$

**B.2** $I_2$

$I_2$ is given by

$$\int_{-\infty}^{\infty} e^{-\alpha \sum_{m=1}^{c}(t-\tau_m)^2} k^{(G)}(t - \boldsymbol{\tau}, \mathbf{z}_j^{(G)}) \prod_{k=1}^{c} (u|\mathbf{v}^u)(\tau_k)\mathrm{d}\tau_k$$

when we substitute in our expression for the SE kernel we get that,

$$I_2 = \sigma_G^2 \int_{-\infty}^{\infty} e^{-\alpha \sum_{m=1}^{c}(t-\tau_m)^2} e^{-p_G \sum_l (t-\tau_l-z_{j,l}^{(G)})^2} \prod_{k=1}^{c} (u|\mathbf{v}^u)(\tau_k) \mathrm{d}\tau_k$$

$$= \sigma_G^2 \int_{-\infty}^{\infty} \prod_{m=1}^{c} e^{-\alpha(t-\tau_m)^2} \prod_{l=1}^{c} e^{-p_G(t-\tau_l-z_{j,l}^{(G)})^2} \prod_{k=1}^{c} (u|\mathbf{v}^u)(\tau_k) \mathrm{d}\tau_k$$

$$= \sigma_G^2 \prod_{i=1}^{c} \int_{-\infty}^{\infty} e^{-\alpha(t-\tau)^2} e^{-p_G(t-\tau-z_{j,i}^{(G)})^2} (u|\mathbf{v}^u)(\tau) \mathrm{d}\tau.$$

Substituting in the expression for the input process $u$, we get

$$I_2 = \sigma_G^2 \prod_{i=1}^{c} \int_{-\infty}^{\infty} e^{-\alpha(t-\tau)^2} e^{-p_G(t-\tau-z_{j,i}^{(G)})^2} \left( \sum_{m=1}^{N_b} w_m^{(u)} \phi_m^{(u)}(\tau) + \sum_{n=1}^{M^{(u)}} q_n^{(u)} k^{(u)}(\tau, z_n^{(u)}) \right) \mathrm{d}\tau$$

$$= \sigma_G^2 \prod_{i=1}^{c} \left[ \sqrt{\frac{2}{N_b}} \sum_{m=1}^{N_b} w_m^{(u)} \int_{-\infty}^{\infty} e^{-\alpha(t-\tau)^2 - p_G(t-\tau-z_{j,i}^{(G)})^2} \cos(\theta_m^{(u)}\tau + \beta_m^{(u)}) \mathrm{d}\tau \right.$$

$$+ \sigma_u^2 \sum_{n=1}^{M^{(u)}} q_n^{(u)} \int_{-\infty}^{\infty} e^{-\alpha(t-\tau)^2 - p_G(t-\tau-z_{j,i}^{(G)})^2} e^{-p_u(\tau-z_n^{(u)})^2} \mathrm{d}\tau \right]$$

$$= \sigma_G^2 \prod_{i=1}^{c} \left[ \sqrt{\frac{2}{N_b}} \sum_{m=1}^{N_b} w_m^{(u)} I_{2a}(t; \alpha, p_G, z_{j,i}^{(G)}, \theta_m^{(u)}, \beta_m^{(u)}) \right.$$

$$+ \sigma_u^2 \sum_{n=1}^{M^{(u)}} q_n^{(u)} I_{2b}(t; \alpha, p_G, z_{j,i}^{(G)}, p_u, z_n^{(u)}) \right].$$

The explicit forms for $I_{2a}$ and $I_{2b}$ are given by,

$$I_{2a}(t; \alpha, p_1, z_1, \theta_2, \beta_2) = \int_{-\infty}^{\infty} e^{-\alpha(t-\tau)^2 - p_1(t-\tau-z_1)^2} \cos(\theta_2\tau + \beta_2) \mathrm{d}\tau$$

$$= \frac{\sqrt{\pi}}{\sqrt{a+p_1}} e^{-\frac{4ap_1 z_1^2 + \theta_2^2}{4(a+p_1)}} \cos\left( \theta_2 \left( t - \frac{p_1 z_1}{a+p_1} \right) + \beta_2 \right)$$

$$I_{2b}(t; \alpha, p_1, z_1, p_2, z_2) = \int_{-\infty}^{\infty} e^{-\alpha(t-\tau)^2 - p_1(t-\tau-z_1)^2} e^{-p_2(\tau-z_2)^2} \mathrm{d}\tau$$

$$= \frac{\sqrt{\pi}}{\sqrt{a+p_1+p_2}} e^{-\frac{a\left( p_1 z_1^2 + p_2(t-z_2)^2 \right) + p_1 p_2(-t+z_1+z_2)^2}{a+p_1+p_2}}.$$

## B.3 Final expression

Collating $I_1$ and $I_2$, we get that the final expression for an explicit sample from the $c$-th order term is

$$(f_d | \{\mathbf{v}_{d,c}^G\}_{c=1}^C, \mathbf{v}^u)(t) = \sum_{c=1}^{C} I_{c,d},$$

with

$$
\begin{aligned}
I_{c,d} = \sum_{i=1}^{N_b} w_i^{(G)} \Bigg( & \frac{1}{2}\sqrt{\frac{2}{N_b}} \Big( e^{i\beta_i^{(G)}} \prod_{j=1}^{c} \Big[ \sqrt{\frac{2}{N_b}} \sum_{m=1}^{N_b} w_m^{(u)} I_{1a}(t;\alpha,\theta_{i,j}^{(G)},\theta_m^{(u)},\beta_m^{(u)}) \\
& + \sigma_u^2 \sum_{n=1}^{M^{(u)}} q_n^{(u)} I_{1b}(t;\alpha,\theta_{i,j}^{(G)},p_u,z_n^{(u)}) \Big] \\
& + e^{-i\beta_i^{(G)}} \prod_{j=1}^{c} \Big[ \sqrt{\frac{2}{N_b}} \sum_{m=1}^{N_b} w_m^{(u)} I_{1a}(t;\alpha,-\theta_{i,j}^{(G)},-\theta_m^{(u)},\beta_m^{(u)}) \\
& + \sigma_u^2 \sum_{n=1}^{M^{(u)}} q_n^{(u)} I_{1b}(t;\alpha,-\theta_{i,j}^{(G)},-p_u,z_n^{(u)}) \Big] \Big) \Bigg) \\
+ \sum_{j=1}^{M^{(G)}} q_j^{(G)} \Bigg( & \sigma_G^2 \prod_{i=1}^{c} \Big[ \sqrt{\frac{2}{N_b}} \sum_{m=1}^{N_b} w_m^{(u)} I_{2a}(t;\alpha,p_G,z_{i,j}^{(G)},\theta_m^{(u)},\beta_m^{(u)}) \\
& + \sigma_u^2 \sum_{n=1}^{M^{(u)}} q_n^{(u)} I_{2b}(t;\alpha,p_G,z_{i,j}^{(G)},p_u,z_n^{(u)}) \Big] \Bigg).
\end{aligned}
$$

## C  Derivation of variational lower bound

Recall that the true joint distribution is given by

$$
p(\{\mathbf{y}_d\}_{d=1}^{D}, \{G_{d,c},\mathbf{v}_{d,c}^G\}_{c,d=1}^{C,D}, u, \mathbf{v}^u) =
$$
$$
\prod_{d,i=1}^{D,N_d} p(y_{d,i}|f_d(t_{d,i})) \prod_{d,c=1}^{D,C} p(G_{d,c}|\mathbf{v}_{d,c}^G)p(\mathbf{v}_{d,c}^G)p(u|\mathbf{v}^u)p(\mathbf{v}^u),
$$

and the variational distribution by

$$
q(\{G_{d,c},\mathbf{v}_{d,c}^G\}_{c,d=1}^{C,D}, u, \mathbf{v}^u) = \prod_{d,c=1}^{D,C} p(G_{d,c}|\mathbf{v}_{d,c}^G)q(\mathbf{v}_{d,c}^G)p(u|\mathbf{v}^u)q(\mathbf{v}^u).
$$

The variational lower bound is the KL divergence between the variational posterior and true posterior, given by

$$
\mathcal{F} = \int q(\{G_{d,c},\mathbf{v}_{d,c}^G\}_{c,d=1}^{C,D}, u, \mathbf{v}^u) \log \frac{p(\{\mathbf{y}_d\}_{d=1}^{D}, \{G_{d,c},\mathbf{v}_{d,c}^G\}_{c,d=1}^{C,D}, u, \mathbf{v}^u)}{q(\{G_{d,c},\mathbf{v}_{d,c}^G\}_{c,d=1}^{C,D}, u, \mathbf{v}^u)} dS,
$$

where $dS$ represents the integral over all inducing points, all VK GPs and input GP. Substituting the expressions for each distribution we obtain,

$$\mathcal{F} = \int q(\{G_{d,c}, \mathbf{v}_{d,c}^G\}_{c,d=1}^{C,D}, u, \mathbf{v}^u)$$

$$\times \log \frac{\prod_{d,i=1}^{D,N_d} p(y_{d,i}|f_d(t_{d,i})) \prod_{d,c=1}^{D,C} \cancel{p(G_{d,c}|\mathbf{v}_{d,c}^G)} p(\mathbf{v}_{d,c}^G) \cancel{p(u|\mathbf{v}_u)} p(\mathbf{v}_u)}{\prod_{d,c=1}^{D,C} \cancel{p(G_{d,c}|\mathbf{v}_{d,c}^G)} q(\mathbf{v}_{d,c}^G) \cancel{p(u|\mathbf{v}_u)} q(\mathbf{v}_u)} dS$$

$$= \int \prod_{d,c=1}^{D,C} p(G_{d,c}|\mathbf{v}_{d,c}^G) q(\mathbf{v}_{d,c}^G) p(u|\mathbf{v}^u) q(\mathbf{v}^u)$$

$$\times \log \frac{\prod_{d,i=1}^{D,N_d} p(y_{d,i}|f_d(t_{d,i})) \prod_{d,c=1}^{D,C} p(\mathbf{v}_{d,c}^G) p(\mathbf{v}_u)}{\prod_{d,c=1}^{D,C} q(\mathbf{v}_{d,c}^G) q(\mathbf{v}_u)} dS$$

$$= \int \prod_{d,c=1}^{D,C} p(G_{d,c}|\mathbf{v}_{d,c}^G) q(\mathbf{v}_{d,c}^G) p(u|\mathbf{v}^u) q(\mathbf{v}^u) \log \prod_{d,i=1}^{D,N_d} p(y_{d,i}|f_d(t_{d,i})) dS$$

$$- \sum_{c,d=1}^{C,D} \mathrm{KL}[q(\mathbf{v}_{d,c}^G)||p(\mathbf{v}_{d,c}^G)] - \mathrm{KL}[q(\mathbf{v}_u)||p(\mathbf{v}_u)]$$

$$= \mathbb{E}_{q(\{G_{d,c}, \mathbf{v}_{d,c}^G\}_{c,d=1}^{C,D}, u, \mathbf{v}^u)} \Big[ \prod_{d,i=1}^{D,N_d} p(y_{d,i}|f_d(t_{d,i})) \Big]$$

$$- \sum_{c,d=1}^{C,D} \mathrm{KL}[q(\mathbf{v}_{d,c}^G)||q(\mathbf{v}_{d,c}^G)] - \mathrm{KL}[q(\mathbf{v}_u)||p(\mathbf{v}_u)]$$

$$= \sum_{d,i=1}^{D,N_d} \mathbb{E}_q[p(y_{d,i}|f_d(t_{d,i}))] - \sum_{c,d=1}^{C,D} \mathrm{KL}[q(\mathbf{v}_{d,c}^G)||p(\mathbf{v}_{d,c}^G)] - \mathrm{KL}[q(\mathbf{v}_u)||p(\mathbf{v}_u)]$$

where we have used the additive property of the KL divergence. The KL terms for the inducing point distributions are between multivariate Gaussians, with $p(\mathbf{v}_{d,c}^G) = \mathcal{N}(0, \mathbf{K}_{d,c}^G)$ and $p(\mathbf{v}^u) = \mathcal{N}(0, \mathbf{K}^u)$, where the $\mathbf{K}$'s represent the covariance matrices, and $q(\mathbf{v}_{d,c}^G) = \mathcal{N}(\boldsymbol{\mu}_{d,c}^G, \boldsymbol{\Sigma}_{d,c}^G)$ and $q(\mathbf{v}^u) = \mathcal{N}(\boldsymbol{\mu}^u, \boldsymbol{\Sigma}^u)$, where the $\boldsymbol{\mu}$'s and $\boldsymbol{\Sigma}$'s represent variational parameters. Using the general result from Rasmussen and Williams [2], we obtain

$$\mathrm{KL}[q(\mathbf{v}_{d,c}^G)||p(\mathbf{v}_{d,c}^G)] = \frac{1}{2} \log |\mathbf{K}_{d,c}^G (\boldsymbol{\Sigma}_{d,c}^G)^{-1}| + \frac{1}{2} \mathrm{tr}[\mathbf{K}_{d,c}^G(\boldsymbol{\mu}_{d,c}^G(\boldsymbol{\mu}_{d,c}^G)^\top + \boldsymbol{\Sigma}_{d,c}^G - \mathbf{K}_{d,c}^G)],$$

and

$$\mathrm{KL}[q(\mathbf{v}^u)||p(\mathbf{v}^u)] = \frac{1}{2} \log |\mathbf{K}^u (\boldsymbol{\Sigma}^u)^{-1}| + \frac{1}{2} \mathrm{tr}[\mathbf{K}^u(\boldsymbol{\mu}^u(\boldsymbol{\mu}^u)^\top + \boldsymbol{\Sigma}^u - \mathbf{K}^u)].$$

## D  Volterra kernel recovery experiment

In this appendix we present an additional synthetic data experiment which allows us to illustrate the ability of the model to learn the correct Volterra series representation directly from data. We generate 600 data points on a grid in the range $[-2, 2]$, by sampling from the NVKM with $C = 2$ and a single output, and adding Gaussian noise with $\sigma_y = 0.025$. We then randomly initialise a model with $C = 3$ terms, and fit it to the data. The process is carried out for both the standard and IO variant of the model, where in the standard case, data for $u$ is not provided, and it is considered a latent function, and for the IO case data for $u$ is provided on the same grid as the outputs, also corrupted with Gaussian noise with $\sigma_u = 0.025$.

If the model has correctly identified the order of VKs present in the data, then the third order term should not contribute significantly to the power of the output, since only the terms up to second order are present in the data. After fitting, the third order term contributed only $0.004\%$ of the power of the output in the standard model, and $0.01\%$ in the IO model. This supports the notion that the model is able to correctly identify the order of non-linearity in the data.

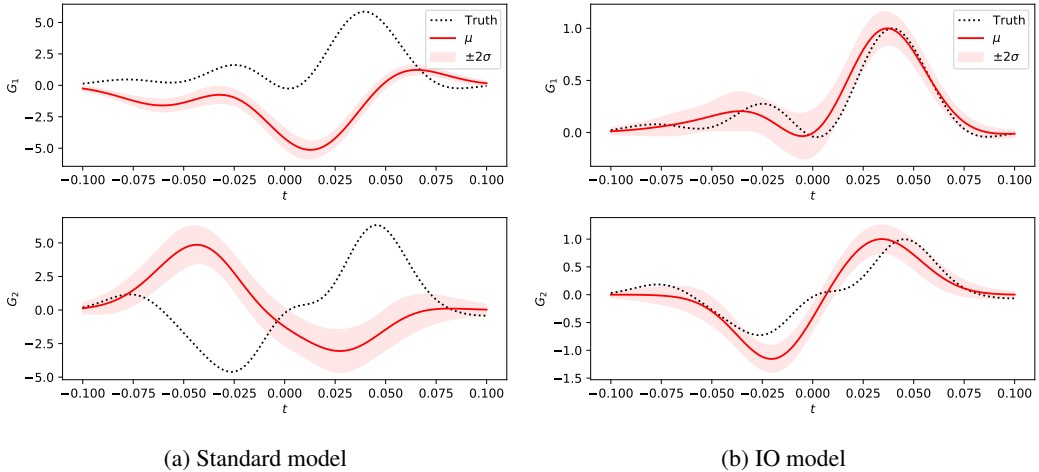

|     |     |
| :-: | :-: |
| (a) Standard model | (b) IO model |

Figure 1: Comparisons of the inferred and true VKs for the recovery experiment, the inferred VKs with $2\sigma$ confidence regions are shown in red solid lines with the true VKs shown in dotted black lines. Plot (a) on the left shows the standard model with first order VK, $G_1(t)$, in the top plot and second order VK, $G_2(t, t)$ in the bottom plot, likewise for the IO model which is shown in (b) on the right. For the second order VK the diagonal is plotted. The third order term is not shown since, as discussed in the text, it does not contribute significantly to the output signal.

In addition to correctly learning the order of the series, it is informative to see if the model can correctly recover the shape of the VKs. A plot of the inferred shapes can be seen in Figure 1. In the standard case, where the input is latent, the true VKs are not exactly recovered. The reason for this is that since the input function is latent, there are many different combinations of input functions and VKs that can produce the same output, so the problem is fundamentally poorly specified. If we look more closely however we can see that many of the qualitative properties of the VKs are recovered. If we were to flip the first order filter along the vertical axis we would see a good agreement, with a large peak followed by a smaller offset peak. Likewise for the second order term, we would see good agreement if we time reversed the inferred VK. In the IO case, we see much better agreement, since the availability of training data for the input function means that the number of combinations of input and VKs that can lead to the correct output is highly constrained. The first order term is recovered very well, with the true VK falling entirely in the confidence region. The recovery in the second order case is less exact but still almost exactly recovers the VK. These results indicate that the model is able to recover the important information about the nature of the non-linearity present in the data.