# OpenReview forum: "Learning Nonparametric Volterra Kernels with Gaussian Processes"
_NeurIPS.cc/2021/Conference — NeurIPS 2021 Poster_

### Official Review · Reviewer_Aww8 · 2021-07-12

**Rating:** 7
**Confidence:** 4

**Summary:**

The paper expands previous work to learn convolutional operators that map a latent (unobserved) GP to observed data. The convolutional operators/kernels are more general as in previous works (non-factorizing and non-homogeneous kernels), and inference in made scalable via a combination of a variation approach and recent results to efficiently sample GPs. Furthermore, the authors provide an extension, where the latent input process is also associated with data.

**Ethical Concerns:**

I found the comment on drones (l. 324ff) a bit out of place. Surely, it is not impossible, that the proposed algorithm could be used in such a malicious way, but the comment seems a bit farfetched, and any technology can be used in such away. Since neither methodology, nor experiments point towards this application, I do not consider this comment as crucial here.

**Limitations And Societal Impact:**

+ l. 153: I don’t think, one can say that the optimal variational distributions are Gaussians. Optimal would be, that they turn out to be Gaussians under the factorizing assumption of Eq. (7). Here, however, I believe it is an assumption that needs to be made.
+ Sec.5.1: While I agree, that the synthetic example is a difficult one, it would be insightful, if an inference example can be shown, that is coming from the generative model. Can the model then correctly identify C? Or does the fact, that the authors choose a decreasing number of inducing points for larger C’s corrupt the model selection?
+ Also for the real experiments, it would be nice to choose an example, where learnt filters can be interpreted in a meaningful way, and potentially demonstrate, why it is important to assume non-homogenous and/or non-factorizing filters, instead of just referring to the likelihood.
+ As discussed, I also consider the fact, that the input process u needs to be univariate the strongest restriction. I would appreciate, in the discussion, how many dimensions are likely to be feasible.
+ I am missing the report of runtimes in the experiments. How does it scale with number of data points, C, etc. How does it compare to the more limited models? Does one pay a high price in terms of computation time for the proposed model?


**Main Review:**

I found the paper interesting and very well written. It is providing some methodological expansion on previous work, such that less assumption on the Gaussian process convolutional models (GPCM) are needed. While the expansions of the model are straightforward, the variational inference of the model needed to utilize some recent findings on GP-sampling to get an efficient approximation of the ELBO. To use the sampling technique from [Wilson et al, 2020] the authors transform a GP with non-stationary kernels, to a stationary one. Then, the nontractable parts of the ELBO can be computed efficiently, and standard stochastic VI methods can be employed.
The experiments seem to be done thoroughly. However, I’d have some suggestion for improvements (see below).
Overall, the paper is not the most innovative one, since novelty mainly in combinations of recent findings and an interesting, but straightforward model extension. But ideas are executed neatly, and I consider it as some solid work, which could be published at this stage.

## Update

I thank the authors for their response. Considering the additional information and experiments, that will be added to the final publication, I am happy to increase my score to 7.

**Time Spent Reviewing:**

5h

---

> ### Author Response · Authors · 2021-08-10
> **Reply to Reviewer Aww8**
>
>
> We thank the reviewer for their extensive comments as well as the kind words about the work, and appreciate the time they put in to review our paper. We hope to address the points they raise in order below:
>
> > l. 153: I don’t think, one can say that the optimal variational distributions are Gaussians. Optimal would be, that they turn out to be Gaussians under the factorizing assumption of Eq. (7). Here, however, I believe it is an assumption that needs to be made.
>
> We appreciate the reviewer pointing out this mistake on our part, and will fix the wording to make it clear the Gaussians are only optimal given the factorizing assumption, not in general, in the updated manuscript.
>
> > Sec.5.1: While I agree, that the synthetic example is a difficult one, it would be insightful, if an inference example can be shown, that is coming from the generative model. Can the model then correctly identify C? Or does the fact, that the authors choose a decreasing number of inducing points for larger C’s corrupt the model selection?
>
> Relating to the reviewer's request for an additional inference example, we have carried out an additional experiment like the reviewer describes. In the experiment, we generate data from both the regular and IO models with $C=2$, and corrupt it with a small amount of noise. We then randomly initialize the models with $C=3$ and fit to the data. After fitting, the $C=3$ term makes up only $0.004\\%$  of the power of output signal in the standard case, and $0.01\\%$ in the IO case, indicating that the model has correctly learned to ignore the $C=3$ term, since it is not present in the training data. Additionally the model learns to recover the salient features of the Volterra kernels (VKs) in the standard case (i.e. with no data for $u$), and in the IO case recovers the shape almost exactly (the reasons for this will be discussed briefly in point 3. below). We plan to include full details and plots for this experiment in the updated supplemental.
>
> > Also for the real experiments, it would be nice to choose an example, where learnt filters can be interpreted in a meaningful way, and potentially demonstrate, why it is important to assume non-homogenous and/or non-factorizing filters, instead of just referring to the likelihood.
>
> The interpretability of the filters is slightly different in the different model variants. In the IO case, the filters have a much more direct meaning, since data for both $u$ and $f$ is available, the set of filters that can map from $u$ to $f$ is much restricted when compared with the standard model, where $u$ is latent and must also be inferred. In the standard model, there are many different combinations of $u$'s and VKs that can lead to the same output, so they are harder to interpret. The experiment we reference in 2. above showed that the general shape of the VKs is recovered in the standard model, but they can be flipped or inverted relative to the VKs that generated the data, whereas in the IO model, the inferred and generated VKs correspond much more closely. On the point about homogenous and factorizing filters, we are currently working on implementing both these approximations, and plan to investigate more thoroughly what is lost (or gained) in making these approximations as part of future work.
>
> > As discussed, I also consider the fact, that the input process u needs to be univariate the strongest restriction. I would appreciate, in the discussion, how many dimensions are likely to be feasible.
>
> We are not entirely yet sure what form the multidimensional input model will have, but the likely scaling is linear in the input dimension, if we assume the input process has a kernel that factorizes (like the ARD kernel for example). This would mean dimensions in the 10s would be  feasible, but likely not in the 100s. We will add a comment on this to the discussion.
>
> > I am missing the report of runtimes in the experiments. How does it scale with number of data points, C, etc. How does it compare to the more limited models? Does one pay a high price in terms of computation time for the proposed model?
>
> Unfortunately we did not measure exact runtimes for the experiments but we will try to clarify a few points below. The runtimes are of course problem dependent, but the models shown in Figure 3 had full training times of approximately 15, 20, 50 and 75 minutes for the 1st to 4th order models respectively. Note that these numbers could likely be improved a lot, we still consider ourselves beginners with the Jax framework, and left the model to run a bit after they likely converged to be on the safe side. Since our model uses mini-batch subsampling the complexity is constant relative to the data size, and scales cubically with the number of inducing points. What this typically means in practice is that for small problems of ~1000 points, models based on standard GP inference, like the LFM typically run faster, since many less gradient evaluations are required for convergence. For larger problems since exact GP inference is cubic with respect to the data size, the mini-batch methods that we use will be faster. Additionally, there is the scaling related to the number of terms in the series. If each of the $C$ VKs has  $m$ inducing points, then the scaling is $\mathcal{O}(Cm^3)$, but alone this is somewhat misleading since as the dimensionality of the VKs increases, the number of inducing points needed to characterize the space, and have a model that can perform well, grows very quickly, effectively limiting the number of terms that is feasible to about five. The homogenous and factorizing approximations will likely help a lot with this, and as mentioned previously, we are currently working on an implementation we hope to share in future work. The model scales linearly in the number of outputs.
>
> > I found the comment on drones (l. 324ff) a bit out of place. Surely, it is not impossible, that the proposed algorithm could be used in such a malicious way, but the comment seems a bit farfetched, and any technology can be used in such away. Since neither methodology, nor experiments point towards this application, I do not consider this comment as crucial here.
>
> Reflecting on this we now see that it is not a very relevant example and thank the reviewer for pointing it out. As per reviewer aBJf's suggestion, we plan on replacing it with a more appropriate  example related to climate/weather modeling.
>
> We again would like to say that we appreciate the time the reviewer spent considering our work, and thank them for their helpful feedback and comments.

---

### Official Review · Reviewer_aBJf · 2021-07-14

**Rating:** 8
**Confidence:** 4

**Summary:**

The paper places GP priors over the kernels of a Volterra series which is driven by an input process drawn from a GP, which may optionally have associated observations (the IO variant of the model). Sampling and mean field variational inference procedures are described. The method is demonstrated on two real data sets.

**Limitations And Societal Impact:**

# Societal Impacts
Suggest to include some possible impact on the modelling aspect. For example, climate/weather modelling which is actually one of the examples.

**Main Review:**

The paper seemed to be inspired by [2]. However, it moved away from the full GP approximation to directly model the the output function via equation (5). This make the paper more original and significant. Sampling and inference procedure are clearly described.

# COMMENTS
1. It will be worthwhile to see if (Markov chain) Monte Carlo posterior sampling of the model can yield better results than currently reported. This will allow the reader to assess if the limitation is because of the model or because of the approximate inference. Section 3.1 is a good start to build this upon. In the least, a comment on this aspect will be useful for the reader.

2. In the experiments, a number of runs via different random seeds is needed, and the best results are reported (though see point 3).  Does the various runs give drastically different results? What is the main resultant source of randomness: the gradient descent; the Monte Carlo estimate of eq (9); or the sampling procedure in eq (5)?

3. Section 5.3 says the best results from five runs are picked, but Table 3 shows standard deviation across three runs. Please clarify or correct.

4. Figure 5 is not very useful without the actual data also plotted in the test region; and it is not referred to at all the the text. Suggest to remove this, and use the space to expand on how one may interpret the kernels learnt in section 5.2.

5. Another future work could be to move beyond mean field approximation.

Changes after rebuttal
=================
4. Figure 5 does indeed have the actual data plotted. It was not very clear, and I missed it in the first reading. Suggest the authors to make this clearer in the revised version.

**Time Spent Reviewing:**

4

---

> ### Author Response · Authors · 2021-08-10
> **Reply to Reviewer aBJf**
>
>
> We first thank the reviewer for their detailed comments on the work, as well as the time they spent reviewing, it is much appreciated. We hope to address the points raised by the reviewer below:
>
> > 1. It will be worthwhile to see if (Markov chain) Monte Carlo posterior sampling of the model can yield better results than currently reported. This will allow the reader to assess if the limitation is because of the model or because of the approximate inference. Section 3.1 is a good start to build this upon. In the least, a comment on this aspect will be useful for the reader.
>
> We agree with the reviewer that the idea of using MCMC to sample from the posterior is a promising one, however we believe that the implementation may be non-trivial for the NVKM. We see the main difficulty as being with the integrals in the Volterra series, since inducing points are required there for the tractability of sampling. It is possible we could use a method with control variates here but we cannot comment further on specifics, see e.g. Titsias 2008 (full reference at bottom of response). More generally, improving the inference scheme, in particular using a more flexible variational posterior, has been something we have been working on. As the reviewer points out near the end of the review, the mean field approximation we employ is restrictive, and we believe there will likely be performance improvements by, for example, modeling the inducing points for each VK jointly. We will add a comment on the above points to the future work section in the revised manuscript.
>
> > 2. In the experiments, a number of runs via different random seeds is needed, and the best results are reported (though see point 3). Does the various runs give drastically different results? What is the main resultant source of randomness: the gradient descent; the Monte Carlo estimate of eq (9); or the sampling procedure in eq (5)?
>
> There is a reasonable amount of randomness between repeats for the experiments, as can be seen by the standard deviations in Tables 1 and 3. We believe, from empirical evidence when running the experiments, that the randomness primarily comes from the random initialization of the variational parameters, implying that the bound has many different local minima which are converged on in gradient descent. To a lesser extent the randomness also stems from the mini batch subsampling of the data. Random initialization is likely far from optimal for the parameters, and from some correspondence with other researchers who have implemented the GPCM model, they had developed some more sophisticated methods for initialization which helped with performance. These methods are based on first fitting the mean of the variational distribution to the data using weighted least squares, then initializing the covariances to a fraction of their priors (see ref \[7\] page 29 for details). We intend to work on a similar method for our model as part of future work which should help alleviate the randomness somewhat.
>
> > 3. Section 5.3 says the best results from five runs are picked, but Table 3 shows standard deviation across three runs. Please clarify or correct.
>
> We appreciate the reviewer drawing attention to this, we now realize it is a little unclear. The 5 repeats we discuss on line 314 is a sort of crude hyper parameter search which we use to try to find a reasonable value for VK width/range. We select the width that performs best (in terms of NLPD) on the *training* data, and then repeat the fitting of that model 3 times with different random seeds to get an idea of the variation in the training process. The mean and standard deviation of the 3 repeats is what is reported in table 3. This has also drawn our attention to an error on line 314 where "four" should be "three". We will fix that error and more clearly explain the process in the revised manuscript.
>
> > 4. Figure 5 is not very useful without the actual data also plotted in the test region; and it is not referred to at all the the text. Suggest to remove this, and use the space to expand on how one may interpret the kernels learnt in section 5.2.
>
> We believe that there may have been a slight misunderstanding here, possibly due our poor formatting of the graph, or the wording of the caption, but please let us know if not. The test data is shown on the graph in blue, as opposed to the training data in black. We now realize that if the paper is not viewed in color, this distinction would be not clear at all. We have made an updated graph with larger points with different shapes for the test and train, as well as used a much lighter shade for the test data. We do believe the plot adds to the paper, to give the reader an idea of the different types of problems the NVKM can solve, but we agree with the reviewer that the plot should be mentioned in the text, we will amend this. As for the interpretation of the VKs in 5.2, we can think of the first order kernel, as representing the response of the linearized systems, with the higher orders accounting for the nonlinearities. We can see the model is less certain about the higher orders because they are difficult to identify in the small amount of data for training. We can see that the third order term is contributing over a smaller length scale, and is steep, allowing it to capture the nonlinearities around 600s and 2900s. We will add some of the details above to section 5.2.
>
> > 5. Another future work could be to move beyond mean field approximation.
>
> We agree with the reviewer that this is a promising avenue for future work, and we plan to mention it in the revised manuscript.
>
> > Suggest to include some possible impact on the modelling aspect. For example, climate/weather modelling which is actually one of the examples.
>
> We agree that the climate modeling example is more compelling than those we listed, and in line with the suggestions of reviewer Aww8, we plan to remove the drone/aerospace example, and replace it with the climate modeling one.
>
> We again thank the reviewer for their time and energy, and hope we have addressed their useful points and suggestions above.
>
> Extra references
> ----------------
> Titsias, Michalis K., Neil D. Lawrence, and Magnus Rattray. "Efficient Sampling for Gaussian Process Inference using Control Variables." NIPS. 2008.

---

> > ### Comment · Reviewer_aBJf · 2021-08-28
> > **Figure 5**
> >
> > Thanks. I now understand the Figure. Suggest to change the colour scheme to take account of those with blue-green colour blindness. A zoom-in inset to the region with no training data will be also useful.
> >
> > I will keep update my initial review to reflect that this.

---

> > > ### Author Response · Authors · 2021-09-03
> > > **Figure 5**
> > >
> > > We appreciate the clarification from the reviewer, and plan to implement their suggestions for the colour scheme and inset for the plot in the revised manuscript.

---

### Official Review · Reviewer_5eAE · 2021-07-15

**Rating:** 7
**Confidence:** 3

**Summary:**

This paper proposes to model volterra kernels with Gaussian processes and develops an inference algorithm based on variational inference.

**Ethical Concerns:**

no.

**Limitations And Societal Impact:**

yes.

**Main Review:**



1. square integrability of greens functions (line 66).

What is required of G to ensure finite output depends on G. That is, if u is bounded then it is sufficient that G is integrable (BIBO stability).

2. Decaying square exponential (line 100-101).

Square integrability of G is sufficient for Tobar [29] because it is convolved with a white noise signal. In the present situation it is not clear that this is works. For example, take the second term in the sum of eq (2):

int  ( int G_{d,2}(t-s_1,t-s_2) u(s_1) d s_1` )  u(s_2) d s_2

which can be made finite if  u is in L_2 and  int G_{d,2}(t-s_1,t-s_2) u(s_1) d s_1` )   is an L_2 function for every t, but is this ensured by square integrability of G?

3. Typo (Line 157).

Lieber -> Leibler.


4. Number of inducing points (Line 240).

The number of inducing points assigned to the filters appears counterintuitive. You need few points to cover a small space and many points to cover a large space but the number of points assigned to the filters is directly opposite to this?

5. Length scale and noise parameters (line 249).

It is not made clear how the length scales are fixed or how the noise hyperparameters are post-hoc estimated.






**Time Spent Reviewing:**

2.5

---

> ### Author Response · Authors · 2021-08-10
> **Reply to Reviewer 5eAE**
>
> We thank the reviewer for their the detailed comments which we hope to address in order below:
>
> > 1. square integrability of greens functions (line 66)
>
> It was an oversight on our part not to include the requirement that $u$ be bounded for the output to be finite in the paper, and we will correct it in the next version by adding a sentence about BIBO stability. We appreciate the reviewer pointing this out.
>
> > 2. Decaying square exponential (line 100-101).
>
> First, to address the example given by the reviewer, i.e. to show that the integral in the $c=2$ term is finite. As mentioned on line 132, we can write $G_{d,2}(t_1, t_2) = \\exp(-\\alpha(t\_1^2+t\_2^2))G'\_{d,2}(t\_1, t\_2)$ where $G'\_{d,2}$ has the SE covariance instead of the DSE, so
> $$\\int^{\\infty}\_{-\\infty}G\_{d,2}(t-s\_1, t-s\_2)u(s\_1)ds\_1u(s\_2)ds\_2=\\int^{\\infty}\_{-\\infty}e^{-\\alpha((t-s\_1)^2+(t-s\_2)^2)}G'\_{d,2}(t-s\_1, t-s\_2)u(s\_1)ds\_1u(s\_2)ds_2$$
> $$=\int^{\infty}\_{-\infty}e^{-\alpha(t-s\_1)^2}G'\_{d,2}(t-s\_1, t-s\_2)u(s\_1)ds\_1 e^{-\alpha(t-s\_2)^2}u(s\_2)ds\_2.$$ We can now see that the inner integral is finite, since both $u$ and $G'\_{d, 2}$ are bounded, which also implies the whole integral is bounded. In addition to this clarification, in the supplemental we will include a more formal derivation showing that the outputs have finite energy. We will also include in the supplemental a derivation showing the relationship on line 132.
>
> > 4. Number of inducing points (Line 240)
>
> As mentioned briefly on lines 235-240, as the order in the series increases, the dimensionality of the filters increases and so many more inducing points are needed to characterize them. When we say 15, 10, 6, 4 inducing points *per axis* this means $15^1=15$, $10^2=100$, $6^3=216$, $4^4=256$ inducing points total, increasing with the larger amount of space they need to cover, as the reviewer pointed out. We agree this was a bit unclear, and we will update the manuscript with it phrased as we have done here. This effect does limit the order of series that is computationally feasible, so we are working on a separable approximation to the filters which will hopefully be included in future work.
>
> > 5. Length scale and noise parameters (line 249).
>
> The input process length scale is fixed to be equal to the grid spacing between the input inducing points i.e. if there are 100 points in between 0 and 1, the length scale is set to 0.1, and is not changed in the optimization process. The noise is estimated by minimizing the bound after all the other parameters have been fit. That is to say we run the optimization until convergence for the variational parameters, and other hyperparameters, whilst the noise is fixed to a small value. After this, we fix all of the other parameters, and just optimize the bound with respect to the noise until convergence. We will clarify this in the updated manuscript.
>
> We will fix the typo on line 157, and we are happy to fix or clarify any other points the reviewer may raise. We again thank the reviewer for their time.

---

### Official Review · Reviewer_o4oR · 2021-07-21

**Rating:** 6
**Confidence:** 5

**Summary:**

The article proposes a methodology for designing covariances kernels in a nonparametric way using Volterra series. The ideas proposed build on existing works on convolution models, Volterra kernels, SysID and MOGP. The paper substantially differ from previous works and the contribution is validated empirically.

**Limitations And Societal Impact:**

yes

**Main Review:**

The authors deliver a thorough and clear literature review, which makes it easy for the reader to identify the motivation and the contribution of this article. As other works in the last decade, this paper focuses on kernel design for GPs, in particular, this kernel takes a nonparametric perspective, a practice that has done in the past but only by few researchers. The main distinction of this work is (in addition to the nonarametirc kernel construction) its application for multiple channels and the inclusion of nonlinearities

In my opinion, this paper is appropriate for NeurIPS but I would appreciate if the authors referred to the following points

- The authors use a fixed grid of the input and kernel, please elaborate why this does not constitute a discrete/parametric model.

- The authors refer several times to the similarities between the proposed method and GPCM. Is the approximate inference the same for both? (structured VI) or the implementation of GPCM in this paper is "enhanced" wrt to the original one?

- The approaches for nonparametric kernels are known by their non-trivial implementation, perhaps the authors would make their code available so that the public can use their method.

- Please elaborate on the computational complexity of the method, hopefully quantitatively. Due to the involved high-dimensional elements, this might be worth considering.

- The proposed method is designed for non-Gaussian data. Then, in the experiments, it seems that the authors implicitly assume that real-world data is non-Gaussian (it's most likely the case). Why not explain which nonlinearities are modelled by the Volterra approach?



minor:
l77: Voterra -> Volterra


**Time Spent Reviewing:**

2

---

> ### Author Response · Authors · 2021-08-10
> **Reply to Reviewer o4oR**
>
> Firstly, we thank the reviewer for their kind words about the work, and hope to address the points raised in the review, in order, below:
>
> > The authors use a fixed grid of the input and kernel, please elaborate why this does not constitute a discrete/parametric model.
>
> As mentioned briefly in the paper (l 105), it is necessary to introduce a finite dimensional, parametric representation of the GPs in order to allow the computation of the integrals in Equation (2) and make sampling tractable at all. We agree that the fact we are using a parametric approximation to a nonparametric model could be made clearer in the text, and will add clarification to the revised manuscript. As for the placement of inducing points on a grid, we agree this is likely sub optimal, and could be improved by, e.g., optimizing the inputs using the bound. This was not done in the present work for computational reasons and the difficulty of implementation.
>
> > The authors refer several times to the similarities between the proposed method and GPCM. Is the approximate inference the same for both? (structured VI) or the implementation of GPCM in this paper is "enhanced" wrt to the original one?
>
> As discussed in the paper (l 204), the GPCM and the NVKM in the case $C=1$ are very similar, however there are a few differences in both the inference method and model itself. Firstly, regarding differences in the types of models, the GPCM uses white noise for the input function $u$, and uses the inter-domain inducing points method to place the inducing points on a low pass version of the input. By contrast we use the EQ covariance for $u$ and place the inducing points on the process itself in the standard way. As for differences regarding the inference, in the GPCM, the bound is analytically tractable, and classic VI is used, with no mini-batch sub sampling of the data, and no Monte Carlo approximation of the expected log likelihood. By contrast we use DSVI. It is hard to say which inference scheme is more effective, and we do not have grounds to suggest ours is enhanced, empirical evidence from our experiments suggest the NVKM with $C=1$ yields very similar performance to the GPCM. It should be noted that a tractable bound is not available for the NVKM with $C>1$. We are happy to add clarification on any of the above points to the manuscript if the reviewer feels this would be useful.
>
> > The approaches for nonparametric kernels are known by their non-trivial implementation, perhaps the authors would make their code available so that the public can use their method.
>
> We thank the reviewer for their appreciation of the tricky implementation and plan on making the code available on a public repo once the reviewing process is complete. The code was also included in the supplemental.
>
> > Please elaborate on the computational complexity of the method, hopefully quantitatively. Due to the involved high-dimensional elements, this might be worth considering.
>
> Due to the mini-batch subsampling, the model has constant time and memory complexity with respect to the data. As is standard with GPs based on stochastic VI with inducing points, the model scales cubically with the number of inducing points on the input function. If each of the $C$ VKs has $m$ inducing points, the scaling is $\mathcal{O}(Cm^3)$. As the reviewer alludes to, the VKs grow in dimensionality as $C$ increases, meaning a large number of inducing points are needed to properly specify the VK if we desire complex features to be represented, which limits the maximum number of terms possible. We are currently working on implementing a separable approximation which would largely alleviate this problem. The model scales linearly in the number of outputs.
>
> > The proposed method is designed for non-Gaussian data. Then, in the experiments, it seems that the authors implicitly assume that real-world data is non-Gaussian (it's most likely the case). Why not explain which nonlinearities are modelled by the Volterra approach?
>
> While it is difficult to pinpoint exactly which types of nonlinearities are present in real world data, the advantage of the NVKM is that since it encompasses the linear case in the first term, we are in a sense automatically detecting the nonlinearities when optimizing the VK magnitudes as part of the inference process, by setting orders not present in the data to zero. To illustrate this point we carried out an additional experiment in which we generated data from the $C=2$ model, for both the standard and IO variants, and then fit both types of the model to the data using $C=3$. After fitting, the $C=3$ term makes up only $0.004\\%$  of the power of output signal in the standard case, and $0.01\\%$ in the IO case, indicating that the model has correctly learned to ignore the $C=3$ term, since it is not present in the training data. We plan to include full details and plots for this experiment in the updated supplemental. The fact that the higher order terms were not optimized out on the real data experiments lends evidence to the presence of nonlinearities in those data. It should also be noted that the Volterra series is not capable of modeling certain properties of nonlinear systems, for example chaos. We will clarify this in the revised paper.
>
> We will fix the typo on line 77, and again thank the reviewer for the useful and constructive feedback on the work.

---

> > ### Comment · Reviewer_o4oR · 2021-08-24
> > **rebuttal acknowledgement**
> >
> > I thank the authors for their thorough and clear response. I am satisfied with this paper and thus I maintain my recommendation.

---

### Decision · Program_Chairs · 2021-09-27

**Decision:**

Accept (Poster)

**Comment:**

The reviewers found the paper interesting and well written, with a thorough and clear literature review, and methodologically sound. All reviewers recommend acceptance, and the remaining concerns are possible to address in the final camera-ready version. I recommend that the authors carefully go through the manuscript in light of the reviewer comments.